# Predicting Molecular Conformation via Dynamic Graph Score Matching

**Shitong Luo\*[1], Chence Shi\*[2,3], Minkai Xu[2,3], Jian Tang[2,4,5]**
[1]Peking University [2]Mila - Québec AI Institute
[3]Université de Montréal [4]HEC Montréal [5]CIFAR AI Research Chair
`luost@pku.edu.cn` , `chence.shi@umontreal.ca`
`minkai.xu@umontreal.ca` , `jian.tang@hec.ca`

## Abstract

Predicting stable 3D conformations from 2D molecular graphs has been a long-standing challenge in computational chemistry. Recently, machine learning approaches have demonstrated very promising results compared to traditional experimental and physics-based simulation methods. These approaches mainly focus on modeling the local interactions between neighboring atoms on the molecular graphs and overlook the long-range interactions between non-bonded atoms. However, these non-bonded atoms may be proximal to each other in 3D space, and modeling their interactions is of crucial importance to accurately determine molecular conformations, especially for large molecules and multi-molecular complexes. In this paper, we propose a new approach called Dynamic Graph Score Matching (DGSM) for molecular conformation prediction, which models both the local and long-range interactions by dynamically constructing graph structures between atoms according to their spatial proximity during both training and inference. Specifically, the DGSM directly estimates the gradient fields of the logarithm density of atomic coordinates according to the dynamically constructed graphs using score matching methods. The whole framework can be efficiently trained in an end-to-end fashion. Experiments across multiple tasks show that the DGSM outperforms state-of-the-art baselines by a large margin, and it is capable of generating conformations for a broader range of systems such as proteins and multi-molecular complexes.

## 1 Introduction

Graph-based representations of molecules has become prevalent in a variety of tasks such as property prediction [13, 31] and molecule generation [17, 33, 45]. However, a more natural and intrinsic representation of a molecule is its 3D geometry or *conformation*, which represents a molecule as a set of 3D coordinates. The 3D representation of molecules is central to many tasks, such as molecular properties prediction and virtual screening. Nevertheless, determining the conformation of a molecule remains a challenging task — both computational approaches, e.g., molecular dynamics (MD) [9], and experimental approaches, e.g., crystallography, are expensive and time-consuming.

Recently, machine learning approaches have demonstrated promising performance for molecular conformation generation. Pioneering methods such as GRAPHDG [36] and CGCF [43] first predict interatomic distances between bonded atoms and then solve 3D coordinates from the predicted distances via a post-processing algorithm. Very recently, Shi et al. proposed the CONFGF [34], which employs the score matching technique [38] to learn pseudo-forces between bonded atoms and iteratively applies the forces to a randomly initialized 3D structure until convergence. CONFGF gets

---

*Equal contribution. Work was done during Shitong's internship at Mila.

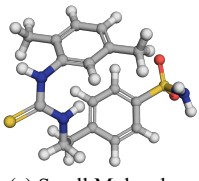 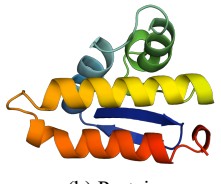 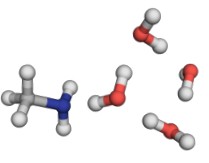

(a) Small Molecule       (b) Protein       (c) Multi-molecular Complex

Figure 1: Three molecular systems where long-range interactions are crucial for their conformations.

rid of the two-stage fashion in prior works and significantly improves the performance. Nevertheless, these approaches have a common major limitation — they mainly focus on modeling the local interactions between bonded atoms defined by the input molecular graphs but fail to capture long-range interactions between non-bonded atoms[1], as they only model distances (or gradients) between bonded atoms. While in molecular mechanics, the potential energy of a molecule that alters conformations can be modeled as a sum of four parts [22]:

$$E = E_{\text{bond}} + E_{\text{angle}} + E_{\text{torsion}} + E_{\text{non-bonded}}, \tag{1}$$

where $E_{\text{bond}}$, $E_{\text{angle}}$, and $E_{\text{torsion}}$ model local interactions between bonded atoms, which are modeled in previous methods [34, 36, 43]. Long-range interactions between non-bonded atoms, denoted as $E_{\text{non-bonded}}$, are also non-trivial, which shape the molecular geometry via non-negligible electrostatic forces or van der Waals forces, etc. For multi-molecular complexes, non-bonded interactions dominate complexes' geometry. An ideal solution to conformation generation should therefore capture both the local and long-range interactions. In Figure 1, we present three typical molecular systems where long-range interactions play a key role in determining their conformations.

To tackle the aforementioned challenge of modeling long-range interactions, in this paper, we propose the Dynamic Graph Score Matching (DGSM) for molecular conformation generation, following the principle of CONFGF [34] that learns the gradients of the logarithm density of atomic coordinates. Instead of relying on the static input molecular graph as existing work, the basic idea is to dynamically construct graph structures between atoms based on their spatial proximity during both training and inference. This allows the model to (1) dynamically learn molecular graph representations with evolved graph structures that take long-range interactions into consideration, and (2) dynamically determine a set of interatomic distances that contribute to gradients of the current atomic coordinates. Specifically, the edges in the dynamic graph consist of two parts. The first part of edges are determined by covalent bonds, which capture local interactions between atoms ($E_{\text{bond}}$, $E_{\text{angle}}$ and $E_{\text{torsion}}$). The second part of edges are determined dynamically by spatial proximity between atoms at each training or sampling step, i.e., two atoms are connected as long as they are proximal, no matter whether they are bonded. Such a strategy is able to effectively capture non-local interactions ($E_{\text{non-bonded}}$) since the magnitude of long-range interactions is inversely correlated with distances between atoms [30]. It remains meanwhile scalable as we avoid connecting all the atom-pairs, which has quadratic complexity. In addition, modeling non-bonded interactions enable the model to sample conformations for multi-molecular complexes, which represent a broader range of problems.

We conduct extensive experiments and compare DGSM against previous state-of-the-art methods on both standard conformation generation and property prediction tasks. Numerical results show that DGSM outperforms previous methods by a clear margin, confirming the benefit of modeling long-range interactions. Besides, to further demonstrate the advantage of DGSM, we bring attention to two more challenging tasks — protein sidechain conformation prediction and multi-molecular complex structure prediction. These two new tasks represent two classes of practical challenges: predicting structures for macro-molecules and multi-molecular complexes.

## 2 Related Work

Prior works on conformation generation mainly rely on molecular dynamics (MD) [9], where new conformations are sequentially generated based on an initial conformation and a physical model for interatomic potentials [25, 27]. Although capable of accurately sampling equilibrium conformations, these methods are computationally intensive, especially for large molecular systems [2, 35], e.g.,

---

[1]Note that the receptive field of graph neural networks is much smaller than the diameter of molecular graphs in large systems, e.g., protein.

proteins. Another category of approaches leverage distance geometry [8] and fix distances between atoms to idealized values heuristically [4], which are much faster but less accurate.

Recently, a variety of deep generative models have been proposed for molecular conformation generation, which strike a good balance between computational efficiency and accuracy. Among these methods, Mansimov et al. [24] first propose a variational autoencoder to directly generate 3D atomic coordinates. Albeit simple, this method fails to model the roto-translation equivariance of molecular conformations, leading to unsatisfactory performance. To preserve roto-translation equivariance, Simm and Hernandez-Lobato [36] and Xu et al. [43] first model the molecular distance geometry and then reconstruct atomic coordinates from generated distances by solving an optimization problem. The state-of-the-art method CONFGF [34] estimates pseudo-forces acting on atoms and generates conformations via Langevin MCMC [42], which bypasses the two-stage fashion in previous works and enhances the performance significantly. Two concurrent works [12, 44] exist which generate conformations in end-to-end fashion via geometry elements assembly and bilevel programming respectively. Recently there has also been attempt to use reinforcement learning for conformation search [14]. Such a method is incapable of modeling bond lengths explicitly, and is fundamentally different from other approaches. To summarize, all of the previous methods focus mainly on modeling the local interactions based on the static input molecular graphs (or augmented graphs by adding auxiliary edges between atoms that are two- and three-hops away) and overlook the long-range non-bonded interactions between atoms. In contrast, our DGSM explicitly models both the local and long-range interactions via dynamic graph score matching and effectively addresses the above issue.

## 3 Preliminaries

### 3.1 Notations and Problem Formulation

**Notations.** Let $\mathcal{G} = \langle \mathcal{V}, \mathcal{E} \rangle$ be a molecular graph, where $\mathcal{V} = \{v_1, v_2, \cdots, v_{|\mathcal{V}|}\}$ is the set of nodes representing atoms, and $\mathcal{E} = \{e_{ij} \mid (i, j) \subseteq \mathcal{V} \times \mathcal{V}\}$ is the set of edges representing inter-atomic bonds in the molecule. Each node $v_i \in \mathcal{V}$ is labeled with atomic attributes, e.g., the element type $Z_i$ and the atomic coordinate $\boldsymbol{r}_i \in \mathbb{R}^3$. Each edge $e_{ij} \in \mathcal{E}$ is labeled with a bond type. The conformation of the molecular graph $\mathcal{G}$ can be represented as a matrix $\boldsymbol{R} \in \mathbb{R}^{|\mathcal{V}| \times 3}$. The distances between all pairs of atoms can be represented as a matrix $\boldsymbol{D} \in \mathbb{R}^{|\mathcal{V}| \times |\mathcal{V}|}$, where $\boldsymbol{D}_{ij} := d_{ij} = \|\boldsymbol{r}_i - \boldsymbol{r}_j\|_2$ denotes the Euclidean distance between the positions of $v_i$ and $v_j$. Following previous work [34, 36, 43], we expand the original molecular graph by adding auxiliary edges between atoms that are second and third neighbors in $\mathcal{G}$ to reduce the degrees of freedom in 3D coordinates.

**Problem Formulation.** Given a molecular graph $\mathcal{G} = \langle \mathcal{V}, \mathcal{E} \rangle$, the task of *molecular conformation generation* is the conditional generation of conformations $\boldsymbol{R} = [\boldsymbol{r}_1; \boldsymbol{r}_2, \cdots; \boldsymbol{r}_{|\mathcal{V}|}] \in \mathbb{R}^{|\mathcal{V}| \times 3}$ based on $\mathcal{G}$, while being able to capture long-range interactions between non-bonded atoms. Note that $\mathcal{G}$ may has multiple connected components, e.g., protein-ligand complexes and multi-molecular complexes.

### 3.2 Score-Based Generative Modeling

Score-based generative modeling [15, 38–40] is a class of generative models that has recently proven effective in a variety of tasks, ranging from image generation [15, 40], audio synthesis [7, 21] to shape generation [6, 23]. For any continuously differentiable probability density $p(\boldsymbol{x})$, we define its *score function* $\boldsymbol{s}(\boldsymbol{x})$ as $\nabla_{\boldsymbol{x}} \log p(\boldsymbol{x})$, i.e., the direction where the logarithm data density grows most rapidly. Score-based generative modeling perturbs the data with different levels of Gaussian noise and jointly estimates the score function of $p(\boldsymbol{x})$ using neural networks. Samples are then generated by sampling from a sequence of decreasing noise levels with Langevin dynamics [42].

Formally, given a data distribution $p_{\text{data}}(\boldsymbol{x})$, let $\{\sigma_i\}_{i=1}^L$ be a sequence of noise levels that satisfies $\sigma_1 > \sigma_2 > \cdots > \sigma_L$ and $\sigma_i/\sigma_{i-1} = \gamma$. Consider a series of noise distributions $p_{\sigma_i}(\tilde{\boldsymbol{x}} \mid \boldsymbol{x}) := \mathcal{N}(\tilde{\boldsymbol{x}}; \boldsymbol{x}, \sigma_i^2 \boldsymbol{I})$, and denote the corresponding perturbed data distribution as $p_{\sigma_i}(\tilde{\boldsymbol{x}}) := \int p_{\sigma_i}(\tilde{\boldsymbol{x}} \mid \boldsymbol{x}) p_{\text{data}}(\boldsymbol{x}) d\boldsymbol{x}$. Song and Ermon [38] propose to jointly approximate the score function of each noise level, denoted by $\boldsymbol{s}_{\boldsymbol{\theta}}(\boldsymbol{x}, \sigma_i)$, with the following objective:

$$\boldsymbol{\theta}^* = \operatorname{argmin}_{\boldsymbol{\theta}} \frac{1}{2L} \sum_{i=1}^L \sigma_i^2 \mathbb{E}_{p_{\text{data}}(\boldsymbol{x})} \mathbb{E}_{p_{\sigma_i}(\tilde{\boldsymbol{x}}|\boldsymbol{x})} \left[ \left\| \boldsymbol{s}_{\boldsymbol{\theta}}(\tilde{\boldsymbol{x}}, \sigma_i) - \nabla_{\tilde{\boldsymbol{x}}} \log p_{\sigma_i}(\tilde{\boldsymbol{x}} \mid \boldsymbol{x}) \right\|_2^2 \right]. \tag{2}$$

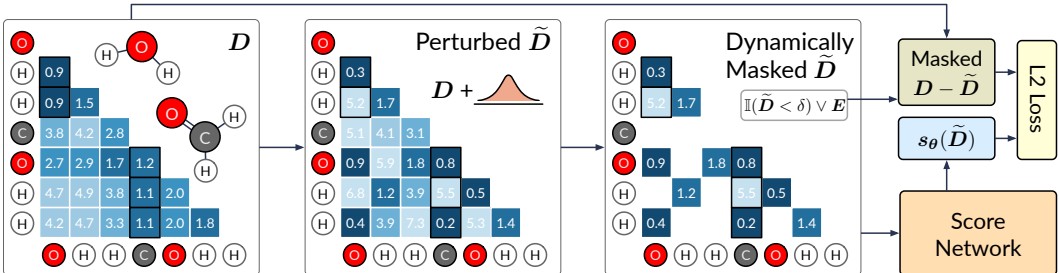

Figure 2: The training procedure of the proposed DGSM. To model long-range interactions, the graph structures are dynamically determined by adding non-bonded edges based on added perturbations at each step. The bonded edges are marked by black border.

Assuming sufficient data and model capacity, the optimal noise conditional score network $s_{\boldsymbol{\theta}^*}(\boldsymbol{x}, \sigma_i)$ matches $\nabla_{\boldsymbol{x}} \log p_{\sigma_i}(\boldsymbol{x})$ almost everywhere [38]. After training score networks, Song and Ermon [38] run annealed Langevin dynamics for each $p_{\sigma_i}(\boldsymbol{x})$ sequentially, where samples from each noise level serve as initializations for Langevin dynamics of the next noise level. Given $\sigma_L$ small enough, the final samples from $p_{\sigma_L}(\boldsymbol{x})$ approximate to samples from $p_{\text{data}}$ under minor conditions.

## 4 Model

Our approach treats the conformation generation as sequentially moving atoms towards high-density regions guided by pseudo-forces, i.e., gradients of atoms. Following Shi et al. [34], we leverage the denoising score matching [38, 41] to approximate the gradients of the logarithm density of atomic coordinates, denoted as $\nabla_{\boldsymbol{R}} \log p(\boldsymbol{R} \mid \mathcal{G})$. To model atomic gradients that are sensitive to both the local and long-range interactions (Eq. 1) and inspired by the fact that long-range interactions decrease rapidly as distances increase, we propose to dynamically construct graph structures with non-bonded edges between atom pairs within a distance based on the current spatial proximity. In this way, we enable the model to effectively capture long-range non-bonded interactions while avoid connecting all atoms, which is computationally expensive. To ensure that the distribution of graph structures during training matches with that during generation, we devise a dynamic graph score matching algorithm, where graph structures are also dynamically determined during training depending on added perturbations. The whole framework is illustrated in Figure 2 and Figure 3. Below we describe the framework of score estimation for Cartesian coordinates in Section 4.1, the dynamic graph score matching algorithm in Section 4.2, and the generation procedure in Section 4.3.

### 4.1 Score Estimation for Cartesian Coordinates

Our goal is to learn the gradients of the logarithm density (score) of atomic coordinates, i.e., $\nabla_{\boldsymbol{R}} \log p(\boldsymbol{R} \mid \mathcal{G})$. Directly parameterizing score networks on absolute Cartesian coordinates with Graph Neural Networks (GNNs) [11, 13, 19, 29] relies on the arbitrary choice of rotation and translation [36, 43], which are non-essential degrees of freedom for effecting conformational changes in molecular systems. Therefore, we explicitly exclude them from the model, by first estimating scores for a set of dynamically determined interatomic distances, and then backpropagating gradients from distances to Cartesian coordinates via differentiation.

Given a molecular graph $\mathcal{G} = \langle \mathcal{V}, \mathcal{E} \rangle$, the probability $p(\boldsymbol{R} \mid \mathcal{G})$ of a conformation $\boldsymbol{R}$ is subject to the Boltzmann distribution and is proportional to $\exp\left(-E(\boldsymbol{R})/k_B T\right)$, where $E(\boldsymbol{R})$ is the conformational energy, $k_B$ is the Boltzmann constant, and $T$ is the temperature. We assume the logarithm density of a conformation, i.e., the negative conformational energy up to a constant, can be parameterized as a function of interatomic distances, conditional on molecular graph $\mathcal{G}$:

$$\log p_{\boldsymbol{\theta}}(\boldsymbol{R} \mid \mathcal{G}) \coloneqq f_{\mathcal{G}}(e_1(\boldsymbol{R}), e_2(\boldsymbol{R}), \cdots, e_K(\boldsymbol{R})), \tag{3}$$

where $\{e_k : \mathbb{R}^{|\mathcal{V}| \times 3} \to \mathbb{R}\}_{k=1}^K$ is a set of functions that calculate $K$ interatomic distances, which are invariant under the rotation and translation of $\boldsymbol{R}$. And $f_{\mathcal{G}} : \mathbb{R}^K \to \mathbb{R}$ is a graph neural network that predicts the negative conformational energy based on distances and the 2D graph representation $\mathcal{G}$. Using interatomic distances for energy prediction is favored in existing literature [16, 20, 31, 34], as it preserves the 3D rotation and translation symmetries of molecular systems. Since

$\{e_k(\boldsymbol{R})\}_{k=1}^K$ are continuously differentiable with respect to the Cartesian coordinates $\boldsymbol{R}$, the gradients of logarithm density of interest, i.e., $\nabla_{\boldsymbol{R}} \log p_{\boldsymbol{\theta}}(\boldsymbol{R} \mid \mathcal{G})$, is interrelated with the logarithm density of each interatomic distance via chain rule:

$$
\begin{aligned}
\forall i, \boldsymbol{s}_{\boldsymbol{\theta}}(\boldsymbol{R})_i &:= \frac{\partial f_{\mathcal{G}}(e_1(\boldsymbol{R}), e_2(\boldsymbol{R}), \cdots, e_K(\boldsymbol{R}))}{\partial \boldsymbol{r}_i} \\
&= \sum_{k=1}^K \frac{\partial f_{\mathcal{G}}(e_1(\boldsymbol{R}), e_2(\boldsymbol{R}), \cdots, e_K(\boldsymbol{R}))}{\partial e_k(\boldsymbol{R})} \cdot \frac{\partial e_k(\boldsymbol{R})}{\partial \boldsymbol{r}_i} \\
&= \sum_{k=1}^K \boldsymbol{s}_{\boldsymbol{\theta}}(e_k(\boldsymbol{R})) \cdot \frac{\partial e_k(\boldsymbol{R})}{\partial \boldsymbol{r}_i},
\end{aligned}
\tag{4}
$$

where $\boldsymbol{s}_{\boldsymbol{\theta}}(\boldsymbol{R})_i$ denotes $\nabla_{\boldsymbol{r}_i} \log p_{\boldsymbol{\theta}}(\boldsymbol{R} \mid \mathcal{G})$, $\boldsymbol{s}_{\boldsymbol{\theta}}(e_k(\boldsymbol{R}))$ denotes $\nabla_{e_k(\boldsymbol{R})} \log p_{\boldsymbol{\theta}}(\boldsymbol{R} \mid \mathcal{G})$, and $\frac{\partial e_k(\boldsymbol{R})}{\partial \boldsymbol{r}_i}$ can be calculated efficiently in closed form (see supplementary material for the full derivation).

Motivated by the above equation, we first train a noise conditional score network to jointly predict the score of interatomic distances, i.e., $\{\boldsymbol{s}_{\boldsymbol{\theta}}(e_k(\boldsymbol{R}), \sigma)\}_{k=1}^K$. After training the noise conditional score network, the gradients of the logarithm density of atomic coordinates, i.e., $\boldsymbol{s}_{\boldsymbol{\theta}}(\boldsymbol{R}, \sigma)$, can be estimated via Eq. 4. We have the following proposition (proof in supplementary material):

**Proposition 1** (Roto-Translation Equivariance)**.** *With the assumption that we can parameterize* $\log p_{\boldsymbol{\theta}}(\boldsymbol{R} \mid \mathcal{G})$ *as a function of interatomic distances, conditional on molecular graph $\mathcal{G}$ (Eq. 3), the score function $\boldsymbol{s}_{\boldsymbol{\theta}}(\boldsymbol{R})$ defined in Eq. 4 is roto-translation equivariant.*

Remarkably, the choice of $\{e_k(\boldsymbol{R})\}_{k=1}^K$ is flexible under this framework and can be carefully designed for specific goals. An ideal set of interatomic distances should capture both the local and long-range interactions between atoms (Eq. 1).

**Proposition 2** (Connection with CONFGF)**.** *The recent proposed* CONFGF *[34] is a special case of our approach, where they only model local distances between the first-order, the second-order, and the third-order neighbors, i.e., $\{e_k\}_{k=1}^K$ map the conformation to a set of distances between bonded atoms. Therefore,* CONFGF *fails to capture long-range interactions between non-bonded atoms.*

## 4.2 Dynamic Graph Score Matching with Noise Conditional Score Networks

In this section, we describe the proposed dynamic graph score matching for interatomic distances, with the goal of modeling both the local and long-range interactions. To ensure that the learned score functions cover all regions with different graph structures, we dynamically construct graph structures with non-bonded edges between atoms during training, based on added perturbations. Following Song and Ermon [38], we train a noise conditional score network to jointly estimate scores for perturbed distributions of a set of dynamically-determined interatomic distances, i.e., $\{\boldsymbol{s}_{\boldsymbol{\theta}}(e_k(\boldsymbol{R}), \sigma)\}_{k=1}^K$, and parameterize the score network with the message passing neural network (MPNN) [13].

**Dynamic Score Matching.** To capture long-range interactions between non-bonded atoms in a molecular system, a naive way is to treat the molecular graph as a fully-connected graph and model the gradients of logarithm density of distances between all pairs of atoms. However, such a practice is computationally expensive especially for large systems, e.g., proteins, and is sometimes unnecessary, e.g., van der Waals interactions decay rapidly as distances increase. As a remedy, we set a cutoff distance and assume each atom only interacts with all atoms within the cutoff distance, ignoring all interactions out of the considered sphere. This is a very popular strategy in computational chemistry that strikes a good balance between efficiency and accuracy [26, 31].

Formally, consider a molecular graph $\mathcal{G} = \langle \mathcal{V}, \mathcal{E} \rangle$ with distances between all pairs of atoms $\boldsymbol{D} \in \mathbb{R}^{|\mathcal{V}| \times |\mathcal{V}|}$ computed from its conformation $\boldsymbol{R} \in \mathbb{R}^{|\mathcal{V}| \times 3}$. For a given noise level $\sigma$, we perturb the distances $\boldsymbol{D}$ with Gaussian noise at each training step on the fly, and then augment the original graph structure with non-bonded edges between atom pairs within a certain threshold distance:

$$
\begin{aligned}
\forall (i, j), \tilde{\boldsymbol{D}}_{ij} &\sim \mathcal{N}(\boldsymbol{D}_{ij}, \sigma^2), \quad \mathcal{E}_\sigma = \mathcal{E} \cup \{e_{ij} \mid \tilde{\boldsymbol{D}}_{ij} < \delta\}, \\
\boldsymbol{d}_\sigma &= \{\boldsymbol{D}_{ij} \mid e_{ij} \in \mathcal{E}_\sigma\}, \quad \tilde{\boldsymbol{d}}_\sigma = \{\tilde{\boldsymbol{D}}_{ij} \mid e_{ij} \in \mathcal{E}_\sigma\},
\end{aligned}
\tag{5}
$$

where $\mathcal{E}_\sigma$ is the constructed graph structure for noise level $\sigma$, and $\delta$ is a hyper-parameter that controls the radius of long-range interactions. We empirically verify that a 10 Å cutoff is sufficient for systems

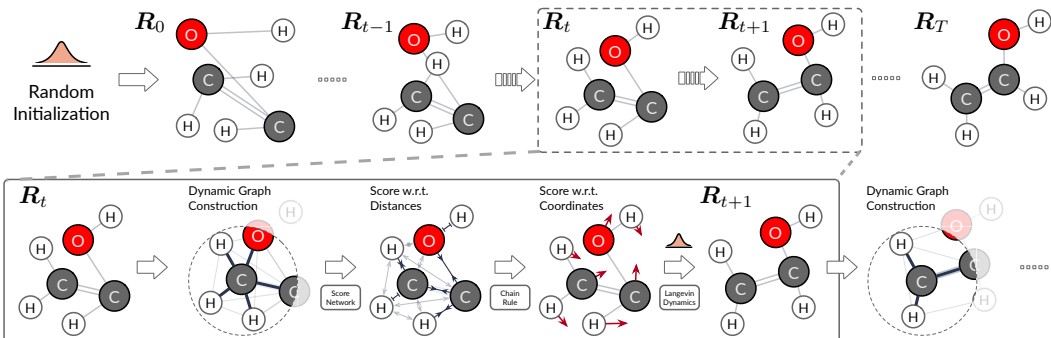

Figure 3: The generation procedure of the proposed DGSM via Langevin dynamics. The graph structure is dynamically constructed at each step of stochastic update based on the current conformation.

we are studying, which is consistent with some experimental results in molecular dynamics [26]. $d_\sigma$ and $\tilde{d}_\sigma$ denote the original and perturbed interatomic distances in augmented graph structure respectively. Hereafter, we omit the subscript for simplicity and use $\mathcal{E}$, $d$, and $\tilde{d}$ instead, assuming all graphs are dynamically constructed during training and sampling.

With the above strategy, the graph structure of a specific molecular graph $\mathcal{G}$ is variadic depending on added perturbation, and all graph structures are possible as long as we sample sufficient enough noise. This will result in (1) a dynamically-determined graph structure for message passing and representation learning, which takes long-range interactions into consideration, and (2) a dynamically-determined set of interatomic distances, i.e., $\{e_k(\boldsymbol{R})\}_{k=1}^K$, for score estimation, which contributes to gradients of atomic coordinates according to Eq. 4. Note that the vanilla implementation of Eq. 5 requires computing all distances between atom pairs. In practice, to avoid quadratic complexity, we pre-filter distant neighbors before adding perturbations for each atom by constructing radius graph with $2\delta$ threshold, and empirically verify that it performs efficiently and effectively.

**Parameterizing with MPNNs.** Let $\{\sigma_i\}_{i=1}^L$ be a sequence of noise levels. Our goal is to learn a noise conditional score network to jointly estimate the scores of all perturbed distance distributions, i.e., $\forall \sigma \in \{\sigma_i\}_{i=1}^L : \boldsymbol{s_\theta}(\tilde{\boldsymbol{d}}, \sigma) \approx \nabla_{\tilde{\boldsymbol{d}}} \log p_\sigma(\tilde{\boldsymbol{d}} \mid \mathcal{G})$, where $p_\sigma(\tilde{\boldsymbol{d}} \mid \mathcal{G}) = \int p(\boldsymbol{d} \mid G) \mathcal{N}(\tilde{\boldsymbol{d}} \mid \boldsymbol{d}, \sigma^2 \boldsymbol{I})$. Following suggestions of Song and Ermon [38], we parameterize the score network with a MPNN as follows (see supplementary material for the full architecture):

$$
\begin{aligned}
\boldsymbol{h}_i^{\text{node}} &= \text{MPNN}(\mathcal{G}, \mathcal{E}, \tilde{\boldsymbol{D}})_i, \quad \forall v_i \in \mathcal{V} \\
\boldsymbol{h}_{ij}^{\text{edge}} &= \text{Concat}(\boldsymbol{h}_i^{\text{node}}, \boldsymbol{h}_j^{\text{node}}), \quad \forall e_{ij} \in \mathcal{E}_\sigma \\
\boldsymbol{s_\theta}(\tilde{\boldsymbol{d}}, \sigma)_{ij} &= \boldsymbol{s_\theta}(\tilde{\boldsymbol{d}})_{ij}/\sigma = \text{MLP}(\boldsymbol{h}_{ij}^{\text{edge}})/\sigma, \quad \forall e_{ij} \in \mathcal{E}_\sigma
\end{aligned}
\tag{6}
$$

where $\{\boldsymbol{h}_i^{\text{node}}\}_{i=1}^{|\mathcal{V}|}$ are node embeddings computed by the MPNN based on the dynamically constructed molecular graph and the perturbed distances, $\{\boldsymbol{h}_{ij}^{\text{edge}}\}$ are embeddings for each edge in $\mathcal{E}$, and $\boldsymbol{s_\theta}(\tilde{\boldsymbol{d}}, \sigma)_{ij}$ is the predicted score for interatomic distance $\tilde{\boldsymbol{D}}_{ij}$ ($e_{ij} \in \mathcal{E}$). The noise conditional network can be jointly optimized with the following objective [38]:

$$
\boldsymbol{\theta}^* = \text{argmin}_{\boldsymbol{\theta}} \frac{1}{2L} \sum_{i=1}^L \sigma_i^2 \mathbb{E}_{p(\boldsymbol{d}|\mathcal{G})} \mathbb{E}_{p_{\sigma_i}(\tilde{\boldsymbol{d}}|\boldsymbol{d},\mathcal{G})} \left[ \left\| \frac{\boldsymbol{s_\theta}(\tilde{\boldsymbol{d}})}{\sigma_i} + \frac{\tilde{\boldsymbol{d}} - \boldsymbol{d}}{\sigma_i^2} \right\|_2^2 \right],
\tag{7}
$$

where all expectations can be efficiently estimated using empirical averages.

### 4.3 Generation

After training the noise conditional score network, molecular conformations can be generated via annealed Langevin dynamics [38], guided by the gradients of atomic coordinates. The gradients can be computed via Eq. 4 based on dynamically constructed graph structures at each step of stochastic update, which allows model to effectively capture both the local and long-range interactions that contribute to the atomic gradients. Formally, given a molecular graph $\mathcal{G}$, we first sample an initial conformation $\boldsymbol{R}_0$ from a fixed prior distribution. We here take the prior distribution as a standard

Gaussian $\mathcal{N}(\boldsymbol{R}_0 \mid \boldsymbol{0}, \boldsymbol{I})$. Then, we update the conformation by running $T$ steps of Langevin dynamic to get a sample from each noise conditional score network $\boldsymbol{s_\theta}(\boldsymbol{R}, \sigma_i)$ sequentially with a special step size schedule $\alpha_i = \varepsilon \cdot \sigma_i^2 / \sigma_L^2$. Samples from each noise level are used to initialize Langevin dynamics for the next noise level. At each sampling step $t$, we first construct graph structures with non-bonded edges within a given distance $\delta$ based on the current pairwise distances $\boldsymbol{D}_{t-1}$ computed from $\boldsymbol{R}_{t-1}$, and then get a set of interatomic distances $\boldsymbol{d}_{t-1}$ for score estimation:

$$\mathcal{E}_{t-1} = \mathcal{E} \cup \{e_{ij} \mid \boldsymbol{D}_{t-1,ij} < \delta\}, \quad \boldsymbol{d}_{t-1} = \{\boldsymbol{D}_{t-1,ij} \mid e_{ij} \in \mathcal{E}_{t-1}\}. \tag{8}$$

The conformation is then updated using the gradient information from the score network (Eq. 4). We provide the pseudo-code in Algorithm 1.

## 5 Experiments

Following previous works [34, 36, 43] on conformation generation, we evaluate the proposed DGSM using the following two standard tasks: **Conformation Generation** (Section 5.1), and **Property Prediction** (Section 5.2). To further demonstrate DGSM's capability of modeling long-range interactions, we evaluate it on two more challenging benchmark tasks: **Protein Sidechain Conformation Generation** and **Multi-molecular Complex Conformation Generation** (Section 5.3). We describe experimental setups in task-specific sections.

---

**Algorithm 1** Annealed Langevin dynamics [38]

**Require:** $\mathcal{G} = \langle \mathcal{V}, \mathcal{E} \rangle, \{\sigma_i\}_{i=1}^L, \delta, \varepsilon, T.$
1: Initialize conformation $\boldsymbol{R}_0$
2: **for** $i \leftarrow 1$ to $L$ **do**
3:    $\alpha_i \leftarrow \varepsilon \cdot \sigma_i^2 / \sigma_L^2$         ▷ $\alpha_i$ is the step size.
4:    **for** $t \leftarrow 1$ to $T$ **do**
5:       $\mathcal{E}_{t-1}, \boldsymbol{d}_{t-1} \leftarrow \text{aug}(\mathcal{E}, \boldsymbol{R}_{t-1}, \delta)$   ▷ Eq. 8
6:       $\boldsymbol{s_\theta}(\boldsymbol{R}_{t-1}, \sigma_i) \leftarrow \text{get}(\boldsymbol{s_\theta}(\boldsymbol{d}_{t-1}, \sigma_i))$ ▷ Eq. 4
7:       Draw $\boldsymbol{z}_t \sim \mathcal{N}(\boldsymbol{0}, \boldsymbol{I})$
8:       $\boldsymbol{R}_t \leftarrow \boldsymbol{R}_{t-1} + \alpha_i \boldsymbol{s_\theta}(\boldsymbol{R}_{t-1}, \sigma_i) + \sqrt{2\alpha_i} \boldsymbol{z}_t$
9:    **end for**
10:   $\boldsymbol{R}_0 \leftarrow \boldsymbol{R}_T$
11: **end for**
**Return:** Generated conformation $\boldsymbol{R}_T$.

---

### 5.1 Conformation Generation

**Setup.** This task evaluates the model's capability to generate stable molecular conformations by measuring both accuracy and diversity of generated conformations. Following previous works [34, 43], we use the GEOM-QM9 and GEOM-Drugs [1] datasets for this task. We use the train-test split provided by [34]. The train splits of GEOM-QM9 and GEOM-Drugs both contain 40,000 molecules, each with 5 conformations for training, or 200,000 conformations in total. The test split of GEOM-QM9 contains 200 molecules with 22,408 conformations, and the test split of GEOM-Drugs contains 200 molecules with 14,324 conformations.

We compare DGSM against 5 state-of-the-art baselines: RDKIT [28], CVGAE [24], GRAPHDG [36], CGCF [43] and CONFGF [34]. For each molecule in the test set, we sample twice as many conformations as its reference conformations. We use the matching score (MAT) to measure the accuracy of generated conformations, and coverage score (COV) to measure the diversity following [34, 43]. Both metrics are based on Root Mean Squared Deviations (RMSD) between molecules, taking symmetries into account (see supplementary material for the details of metrics).

**Results.** We report the mean and median COV and MAT scores over all the molecules in the test split on GEOM-QM9 and GEOM-Drugs datasets. As shown in Table 1, DGSM consistently outperforms

Table 1: COV and MAT scores on GEOM-QM9 and GEOM-Drugs datasets. The threshold $\delta$ of COV score is 0.5Å for GEOM-QM9 and 1.25Å for GEOM-Drugs following Xu et al. [43]. ($\uparrow$): the higher the better. ($\downarrow$): the lower the better.

| | GEOM-QM9 | | | | GEOM-Drugs | | | |
|---|---|---|---|---|---|---|---|---|
| | COV (%, $\uparrow$) | | MAT (Å, $\downarrow$) | | COV (%, $\uparrow$) | | MAT (Å, $\downarrow$) | |
| Method | Mean | Median | Mean | Median | Mean | Median | Mean | Median |
| RDKIT [28] | 83.26 | 90.78 | 0.3447 | 0.2935 | 60.91 | 65.70 | 1.2026 | 1.1252 |
| CVGAE [24] | 0.09 | 0.00 | 1.6713 | 1.6088 | 0.00 | 0.00 | 3.0702 | 2.9937 |
| GRAPHDG [36] | 73.33 | 84.21 | 0.4245 | 0.3973 | 8.27 | 0.00 | 1.9722 | 1.9845 |
| CGCF [43] | 77.52 | 80.40 | 0.4206 | 0.3903 | 54.19 | 56.35 | 1.2575 | 1.2356 |
| CONFGF [34] | 88.49 | 94.13 | 0.2673 | 0.2685 | 62.15 | 70.93 | 1.1629 | 1.1596 |
| **DGSM** | **91.49** | **95.92** | **0.2139** | **0.2137** | **78.73** | **94.39** | **1.0154** | **0.9980** |

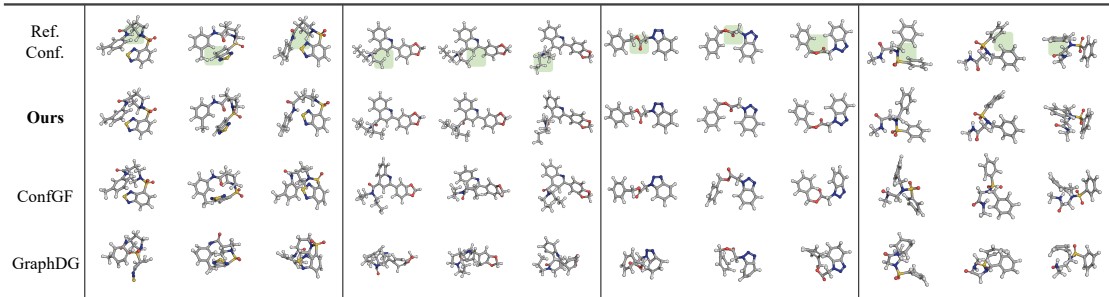

Figure 4: Examples of conformations generated by different models based on four random molecular graphs from the test set of GEOM-Drugs. We present three reference conformations for each molecule, and visualize the best-aligned conformations generated by each method. Areas where long-range interactions should be modeled are highlighted in green.

all the baselines. Notably, both DGSM and CONFGF are score-based models, but DGSM achieves better performance. The difference between them is that DGSM successfully takes long-range interactions into consideration via dynamic graph score matching. This confirms the significant benefit of modeling long-range interactions. We present several conformations generated by different approaches in Figure 4, which shows that DGSM successfully captures the long-range interactions in highlighted areas while the other baselines fail, resulting in distorted structures in those areas.

## 5.2 Property Prediction

**Setup.** This task demonstrates how generative models for molecular conformations can be applied to property prediction as a downstream task. It also provides an assessment on the quality of generated conformations in a different light. We estimate the *ensemble properties* [1] of a molecular graph by aggregating its conformational properties following [34]. In specific, we first use the models

Table 2: Mean absolute errors (MAE) of predicted ensemble properties in eV.

| Method | $\overline{E}$ | $E_{\min}$ | $\overline{\Delta\epsilon}$ | $\Delta\epsilon_{\min}$ | $\Delta\epsilon_{\max}$ |
|---|---|---|---|---|---|
| RDKIT | **0.9233** | 0.6585 | 0.3698 | **0.8021** | 0.2359 |
| GRAPHDG | 9.1027 | 0.8882 | 1.7973 | 4.1743 | 0.4776 |
| CGCF | 28.9661 | 2.8410 | 2.8356 | 10.6361 | 0.5954 |
| CONFGF | 2.7886 | 0.1765 | 0.4688 | 2.1843 | 0.1433 |
| DGSM | 1.0313 | **0.0761** | **0.1963** | 1.1811 | **0.1271** |

to generate 50 conformations for each molecular graph in a subset of GEOM-QM9 [34], and use PSI4 [37], a quantum chemical toolkit, to calculate each conformation's energy and HOMO-LUMO gap. Then, we calculate average energy $\overline{E}$, lowest energy $E_{\min}$, average gap $\overline{\Delta\epsilon}$, minimum gap $\Delta\epsilon_{\min}$, and maximum gap $\Delta\epsilon_{\max}$ from the conformational energy and gap. We evaluate the accuracy of estimated ensemble property by measuring their mean absolute errors (MAE) to the ground truth values. CVGAE is excluded in this task as its performance is poor, which is also reported in [36, 34].

**Results.** Table 2 shows that DGSM outperforms other machine learning-based methods by a clear margin. DGSM's estimation of average energy $\overline{E}$ and minimum gap $\Delta\epsilon_{\min}$ is close to RDKIT but still outperforms the most competitive ML-based method CONFGF. The calculation of conformational energy is highly sensitive to changes in geometry — even a subtle deviation in bond lengths leads to significant energy change [36]. Therefore, the superior performance of DGSM indicates that it generates much more accurate conformations than other methods, leading to more accurate property estimation. This validates again the effectiveness of modeling long-range interactions.

## 5.3 Large Molecule and Multi-molecular Modeling

**Protein Sidechain Conformation** This task is to predict protein sidechain conformations based on its backbone structures. Compared to conventional molecular conformations generation in previous sections, the main challenge of this task is two-fold: (1) large number of atoms, which prohibits constructing complete graphs that grow quadratically to model long-range interactions.

Table 3: RMSD of different approaches on sidechain conformation generation.

| Method | RMSD | |
|---|---|---|
| | Mean (Å) | Min (Å) |
| CONFGF | 3.38 | 3.11 |
| **DGSM** | **2.85** (↓ 15.7%) | **2.61** (↓ 16.1%) |

(2) covalent bonds are sparse, which limits the power of the edge augmentation techniques in previous work. DGSM tackles these two challenges via dynamic graph score matching as introduced.

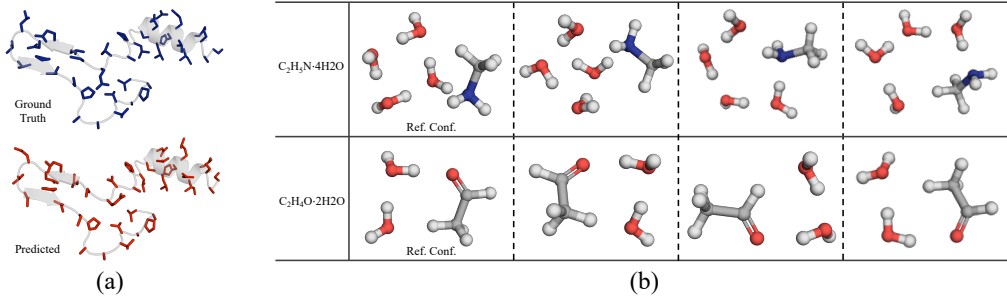

(a)                           (b)

Figure 5: (a) An example of the generated protein sidechain conformation with atomic-level coordinates. The ground-truth sidechain (blue) and the generated sidechain (red) are highlighted. (b) Conformations of two multi-molecular complexes generated by DGSM.

We use the SidechainNet [18] dataset for this task and follow the official train-test splits. We compare DGSM with the state-of-the-art conformation generation model CONFGF. Despite that there are some machine learning-based methods specialized in protein sidechain structure prediction [10], they are built upon rotamer libraries [5], which incorporates a lot of domain knowledge. Thus, our method is not comparable with them. The main purpose of this task is to justify the effectiveness of DGSM for large molecules. For each protein, we generate 5 sidechain conformations with different initialization, and calculate the mean and min RMSD between the ground-truth conformation and the generated conformations. We report the overall mean and min RMSD scores by averaging scores of each protein in the test set in in Table 3, which shows that DGSM achieves better performance than previous state-of-the-art model. We also present an example in Figure 5(a), and we can see that the predicted conformation is consistent with the ground truth in major parts.

**Multi-molecular Complex Conformation**     This task is to predict conformations for multi-molecular complexes. A multi-molecular complex is made up of multiple molecules and there is no covalent bonds between them. Long-range interactions dominate the structure of multi-molecular complexes. The purpose of this task is to demonstrate DGSM's potential application to a broader range of problems and provide a novel benchmark for conformation generation. We use the quantum chemical software xtb [3] to construct a dataset consisting of 24 water-organic complexes each with several hundreds of conformations, and leave out 4 complexes for testing (see supplementary material for details). We do not report RMSD-based metrics such as COV and MAT because the structures of multi-molecular complexes are highly flexible. Two set of generated examples are presented in Figure 5(b). We observe that water molecules are placed

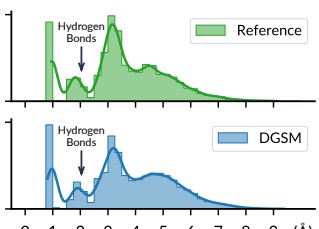

Figure 6: The distribution of Hydrogen-Oxygen distances. The first peak from the left is covalent bonds and the second peak is hydrogen bonds.

regularly around the solute organic molecule. Notably, hydrogen bonds (between water and the solute, and between water and water) are formed correctly. This can also be evidenced in the histogram of Hydrogen-Oxygen distances (Figure 6), where there is a peak between 1.5Å and 2.5Å, i.e., the range of hydrogen bond length between Hydrogen and Oxygen.

# 6   Conclusion and Future Work

We propose DGSM, a novel score-based approach for generating equilibrium molecular conformations. DGSM is capable of modeling both the local and long-range interactions in molecular systems, by dynamically constructing graph structures based on spatial proximity between atoms during both training and inference. We also devise a dynamic graph score matching algorithm to effectively estimate atomic gradients, where graph structures are dynamically determined depending on added perturbations. Extensive experiments over two standard tasks and two original tasks show that DGSM outperforms the state-of-the-art method by a large margin, confirming the significant benefit of modeling long-range interactions. In the future, we plan to apply our approach to the more challenging problem of protein structure prediction.

## Acknowledgments and Disclosure of Funding

We would like to thank all the reviewers for the insightful comments. This project is supported by the Natural Sciences and Engineering Research Council (NSERC) Discovery Grant, the Canada CIFAR AI Chair Program, collaboration grants between Microsoft Research and Mila, Samsung Electronics Co., Ldt., Amazon Faculty Research Award, Tencent AI Lab Rhino-Bird Gift Fund and a NRC Collaborative R&D Project (AI4D-CORE-06). This project was also partially funded by IVADO Fundamental Research Project grant PRF-2019-3583139727.

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
