# Supplementary Material

## A Proof of Proposition 1

### A.1 Full Derivation of Equation 4

Given a molecular graph $\mathcal{G} = \langle \mathcal{V}, \mathcal{E} \rangle$, let $\boldsymbol{R} = [\boldsymbol{r}_1; \boldsymbol{r}_2, \cdots; \boldsymbol{r}_{|\mathcal{V}|}] \in \mathbb{R}^{|\mathcal{V}| \times 3}$ denote its conformation. Let $\{e_k : \mathbb{R}^{|\mathcal{V}| \times 3} \to \mathbb{R}\}_{k=1}^K$ denote a set of functions that calculate $K$ interatomic distances, the gradients of logarithm density of atomic coordinates, i.e., $\nabla_{\boldsymbol{R}} \log p_{\boldsymbol{\theta}}(\boldsymbol{R} \mid \mathcal{G})$, can be calculated using the following equation (see Eq.4):

$$\forall i, \boldsymbol{s}_{\boldsymbol{\theta}}(\boldsymbol{R})_i := \frac{\partial f_{\mathcal{G}}(e_1(\boldsymbol{R}), e_2(\boldsymbol{R}), \cdots, e_K(\boldsymbol{R}))}{\partial \boldsymbol{r}_i} \tag{9}$$

$$= \sum_{k=1}^K \frac{\partial f_{\mathcal{G}}(e_1(\boldsymbol{R}), e_2(\boldsymbol{R}), \cdots, e_K(\boldsymbol{R}))}{\partial e_k(\boldsymbol{R})} \cdot \frac{\partial e_k(\boldsymbol{R})}{\partial \boldsymbol{r}_i} \tag{10}$$

$$= \sum_{k=1}^K \boldsymbol{s}_{\boldsymbol{\theta}}(e_k(\boldsymbol{R})) \cdot \frac{\partial e_k(\boldsymbol{R})}{\partial \boldsymbol{r}_i} \tag{11}$$

$$= \sum_{e_{ij} \in \{e_k\}_{k=1}^K} \boldsymbol{s}_{\boldsymbol{\theta}}(e_{ij}(\boldsymbol{R})) \cdot \frac{\partial e_{ij}(\boldsymbol{R})}{\partial \boldsymbol{r}_i} + \sum_{\substack{i \neq u, i \neq v \\ e_{uv} \in \{e_k\}_{k=1}^K}} \boldsymbol{s}_{\boldsymbol{\theta}}(e_{uv}(\boldsymbol{R})) \cdot \boldsymbol{0} \tag{12}$$

$$= \sum_{e_{ij} \in \{e_k\}_{k=1}^K} \frac{1}{d_{ij}} \cdot \boldsymbol{s}_{\boldsymbol{\theta}}(e_{ij}(\boldsymbol{R})) \cdot (\boldsymbol{r}_i - \boldsymbol{r}_j), \tag{13}$$

where $\boldsymbol{s}_{\boldsymbol{\theta}}(\boldsymbol{R})_i$ denotes $\nabla_{\boldsymbol{r}_i} \log p_{\boldsymbol{\theta}}(\boldsymbol{R} \mid \mathcal{G})$, and $\boldsymbol{s}_{\boldsymbol{\theta}}(e_{ij}(\boldsymbol{R}))$ denotes $\nabla_{e_{ij}(\boldsymbol{R})} \log p_{\boldsymbol{\theta}}(\boldsymbol{R} \mid \mathcal{G})$. We here abuse the notation a little bit and use $e_{ij}$ to denote the interatomic distance between positions of $v_i$ and $v_j$. The term $\frac{\partial e_{uv}(\boldsymbol{R})}{\partial \boldsymbol{r}_i}$ for distances not adjacent to $v_i$, i.e., $i \neq u$ and $i \neq v$, is equal to zero, as these distances are irrelevant to $\boldsymbol{r}_i$. The $\frac{\partial e_{ij}(\boldsymbol{R})}{\partial \boldsymbol{r}_i}$ is equal to $\frac{1}{d_{ij}}(\boldsymbol{r}_i - \boldsymbol{r}_j)$ because $e_{ij}(\boldsymbol{R}) = d_{ij} = \|\boldsymbol{r}_i - \boldsymbol{r}_j\|_2$.

### A.2 Proof of Roto-Translation Equivariance

Let $\mathcal{T}_{\boldsymbol{t}} : \mathbb{R}^{|V| \times 3} \to \mathbb{R}^{|V| \times 3}$ be an arbitrary 3D translation function where $\mathcal{T}_{\boldsymbol{t}}(\boldsymbol{R})_i := \boldsymbol{r}_i + \boldsymbol{t}$, and let $\mathcal{R}_{\boldsymbol{A}} : \mathbb{R}^{|V| \times 3} \to \mathbb{R}^{|V| \times 3}$ be an arbitrary 3D rotation function with a rotation matrix $\boldsymbol{A} \in \mathbb{R}^{3 \times 3}$, i.e., $\mathcal{R}_{\boldsymbol{A}}(\boldsymbol{R})_i := \boldsymbol{A}\boldsymbol{r}_i$. Formally, a score function of Cartesian coordinates $\boldsymbol{s} : \mathbb{R}^{|V| \times 3} \to \mathbb{R}^{|V| \times 3}$ being roto-translation equivariant can be expressed as:

$$\boldsymbol{s} \circ \mathcal{T}_{\boldsymbol{t}} \circ \mathcal{R}_{\boldsymbol{A}}(\boldsymbol{R}) = \mathcal{R}_{\boldsymbol{A}} \circ \boldsymbol{s}(\boldsymbol{R}). \tag{14}$$

*Proof.* Let $\hat{\boldsymbol{R}} := \mathcal{T}_{\boldsymbol{t}} \circ \mathcal{R}_{\boldsymbol{A}}(\boldsymbol{R})$, and therefore $\hat{\boldsymbol{R}}_i = \hat{\boldsymbol{r}}_i = \boldsymbol{A}\boldsymbol{r}_i + \boldsymbol{t}$. Following the strategy in Shi et al. [34] and according to Eq. 13, we have:

$$\forall i, \left( \boldsymbol{s} \circ \mathcal{T}_{\boldsymbol{t}} \circ \mathcal{R}_{\boldsymbol{A}}(\boldsymbol{R}) \right)_i = \boldsymbol{s}(\hat{\boldsymbol{R}})_i \tag{15}$$

$$= \sum_{e_{ij} \in \{e_k\}_{k=1}^K} \frac{1}{\hat{d}_{ij}} \cdot \boldsymbol{s}_{\boldsymbol{\theta}}(e_{ij}(\hat{\boldsymbol{R}})) \cdot (\hat{\boldsymbol{r}}_i - \hat{\boldsymbol{r}}_j) \tag{16}$$

$$= \sum_{e_{ij} \in \{e_k\}_{k=1}^K} \frac{1}{d_{ij}} \cdot \boldsymbol{s}_{\boldsymbol{\theta}}(e_{ij}(\boldsymbol{R})) \cdot ((\boldsymbol{A}\boldsymbol{r}_i + \boldsymbol{t}) - (\boldsymbol{A}\boldsymbol{r}_j + \boldsymbol{t})) \tag{17}$$

$$= \sum_{e_{ij} \in \{e_k\}_{k=1}^K} \frac{1}{d_{ij}} \cdot \boldsymbol{s}_{\boldsymbol{\theta}}(e_{ij}(\boldsymbol{R})) \cdot \boldsymbol{A}(\boldsymbol{r}_i - \boldsymbol{r}_j) \tag{18}$$

$$= A\left( \sum_{e_{ij} \in \{e_k\}_{k=1}^K} \frac{1}{d_{ij}} \cdot s_\theta(e_{ij}(\boldsymbol{R})) \cdot (\boldsymbol{r}_i - \boldsymbol{r}_j) \right) \tag{19}$$

$$= \boldsymbol{A}\boldsymbol{s}(\boldsymbol{R})_i \tag{20}$$

$$= \left( \mathcal{R}_{\boldsymbol{A}} \circ \boldsymbol{s}(\boldsymbol{R}) \right)_i. \tag{21}$$

Here $d_{ij} = \hat{d}_{ij}$ and $e_{ij}(\boldsymbol{R}) = e_{ij}(\hat{\boldsymbol{R}})$ because interatomic distances are invariant under the rotation and translation of $\boldsymbol{R}$. Combining the results in Section A.1 and above, we prove that the score function is roto-translation equivariant. □

# B  Additional Model Details

## B.1  Message Passing Neural Networks

The message passing formula of the MPNN used in DGSM is:

$$h_i^{(\ell+1)} = \sigma\left( \boldsymbol{W}_0^{(\ell)} h_i^{(\ell)} + \sum_{(i,j) \in \mathcal{E}} \boldsymbol{W}_2^{(\ell)}(\boldsymbol{f}^{(\ell)}(d_{ij}, t_{ij}) \odot (\boldsymbol{W}_1^{(\ell)} h_j^{(\ell)})) \right), \tag{22}$$

where $\boldsymbol{f}^{(\ell)}$ is filter network which takes edge length $d_{ij}$ and edge type $t_{ij}$ as input and outputs a weight vector with the same dimension to $(\boldsymbol{W}_1^{(\ell)} h_j^{(\ell)})$, and $\odot$ denotes element-wise multiplication. $\boldsymbol{W}_0^{(\ell)}$, $\boldsymbol{W}_1^{(\ell)}$, and $\boldsymbol{W}_2^{(\ell)}$ are weight matrices that mix the channels of feature vectors. $\sigma$ is the activation function, and we use `ShiftedSoftplus`. The formulation is an extension of continuous filter convolution [32], where we add edge types as an input to the filter network.

## B.2  Hyper-parameters

The hyper-parameters of DGSM for various tasks are summarized in Table 4, including highest noise level $\sigma_1$, lowest noise level $\sigma_L$, number of noise levels $L$, number of steps for each noise level $T$, minimum step size $\epsilon$, cutoff radius for dynamic graph construction $\delta$, training batch size, and number of training iterations. Each model is trained on a single 2080ti GPU.

Table 4: Additional hyperparameters of our DGSM.

| Task | $\sigma_1$ | $\sigma_L$ | $L$ | $T$ | $\epsilon$ | $\delta$ | Batch Size | Train Iter. |
|------|-----------|-----------|-----|-----|-----------|---------|-----------|-------------|
| QM9 | 10 | 0.01 | 50 | 100 | 2.4e-6 | 10Å | 64 | 1M |
| Drugs | 10 | 0.01 | 50 | 100 | 2.4e-6 | 10Å | 32 | 1M |
| Complex | 10 | 0.01 | 50 | 100 | 2.4e-6 | 10Å | 32 | 1M |
| Sidechain | 20 | 0.1 | 100 | 50 | 3.5e-4 | 5Å | 4 | 1M |

# C  Additional Experimental Details

## C.1  Metrics

The coverage score (COV) and matching score (MAT) [43] adopted in the conformation generation experiments are defined as:

$$\text{COV}(S_g, S_r) = \frac{\left| \left\{ \mathbf{R} \in S_r \,|\, \text{RMSD}(\mathbf{R}, \hat{\mathbf{R}}) \leq \tau, \hat{\mathbf{R}} \in S_g \right\} \right|}{|S_r|},$$

$$\text{MAT}(S_g, S_r) = \frac{1}{|S_r|} \sum_{\mathbf{R} \in S_r} \min_{\hat{\mathbf{R}} \in S_g} \text{RMSD}(\mathbf{R}, \hat{\mathbf{R}}), \tag{23}$$

where $S_g$ is the set of generated conformations and $S_r$ is the set of reference conformations. We use the `GetBestRMS` function provided by RDKit[28] to compute the RMSD between two conformations

based on heavy atoms only, following previous works [34, 43]. Note that the RMSD implementation has already taken symmetry into account.

In addition to COV and MAT score, we also report mismatch rates (MIS) [34] in the additional experiment (Section C.4):

$$\text{MIS}(S_g, S_r) = \frac{1}{|S_g|} \left| \left\{ \mathbf{R} \in S_g \,|\, \text{RMSD}(\mathbf{R}, \hat{\mathbf{R}}) > \tau, \forall \hat{\mathbf{R}} \in S_r \right\} \right|, \tag{24}$$

where $\tau$ is the threshold. The mismatch rate is the percentage of generated conformations that are not matched by any reference conformation. Higher mismatch rate indicates more invalid conformations are generated.

Recently, [12] proposed COV-Precision and MAT-Precision scores:

$$\text{COV-P}(S_g, S_r) = \frac{\left| \left\{ \mathbf{R} \in S_g \,|\, \text{RMSD}(\mathbf{R}, \hat{\mathbf{R}}) \leq \tau, \hat{\mathbf{R}} \in S_r \right\} \right|}{|S_g|},$$

$$\text{MAT-P}(S_g, S_r) = \frac{1}{|S_g|} \sum_{\mathbf{R} \in S_g} \min_{\hat{\mathbf{R}} \in S_r} \text{RMSD}(\mathbf{R}, \hat{\mathbf{R}}), \tag{25}$$

We also report these scores for our model and the baselines in Table 5.

## C.2 Multi-Molecular Complex Datasets

The training set for the multi-molecular conformation generation task (Section 5.3) consists of complexes with the form: $X + 4H_2O$, where X is an organic molecules. There are 20 complexes in the training set and 26,303 conformations in total. The SMILES strings of organic molecules appeared in the training set are: `CO`, `CC(O)C`, `CCCCCO`, `OCCC(C)C`, `OCC(C)CC`, `CC(CO)(C)C`, `OC(CC)CC`, `CC(C)C(C)O`, `CCC(C)(C)O`, `COC`, `CCOCC`, `CC(C)(C)OC`, `C1CCOC1`, `O1CCOCC1`, `C=O`, `CCC=O`, `O=CCCC`, `CC(=O)C`, `O=CO`, `CC(O)=O`. The testing set contains 4 complexes: $C_2H_4O \cdot 2\,H_2O$, $C_5H_{12}O \cdot 3\,H_2O$, $CH_5N \cdot 4\,H_2O$, and $C_2H_6O \cdot 5\,H_2O$.

## C.3 COV-Precision and MAT-Precision Scores

We report COV-Precision and MAT-Precision [12] scores as below:

Table 5: COV-Precision and MAT-Precision scores on GEOM-QM9 and GEOM-Drugs datasets. The threshold $\delta$ of COV score is 0.5Å for GEOM-QM9 and 1.25Å for GEOM-Drugs. (↑): the higher the better. (↓): the lower the better.

| Method | GEOM-QM9 | | | | GEOM-Drugs | | | |
| --- | --- | --- | --- | --- | --- | --- | --- | --- |
| | COV-P (%, ↑) | | MAT-P (Å, ↓) | | COV-P (%, ↑) | | MAT-P (Å, ↓) | |
| | Mean | Median | Mean | Median | Mean | Median | Mean | Median |
| GRAPHDG [36] | 43.90 | 35.33 | 0.5809 | 0.5823 | 2.08 | 0.00 | 2.4340 | 2.4100 |
| CGCF [43] | 36.49 | 33.57 | 0.6615 | 0.6427 | 21.68 | 13.72 | 1.8571 | 1.8066 |
| CONFGF [34] | **46.43** | 43.41 | **0.5224** | 0.5124 | 23.42 | 15.52 | 1.7219 | 1.6863 |
| **DGSM** | 44.64 | **43.72** | 0.5369 | **0.5023** | **40.08** | **37.15** | **1.4994** | **1.4496** |

## C.4 Additional COV and MIS Scores

We report COV and MIS score at different thresholds in the tables below. The results show that DGSM consistently outperforms previous methods by a significant margin.

Table 6: COV and MIS scores of different methods on GEOM-QM9 dataset at different thresholds ($\tau$). ($\uparrow$) indicates higher is better. ($\downarrow$) indicates lower is better.

| QM9 | Mean COV (%, ↑) | | | | Median COV (%, ↑) | | | | Mean MIS (%, ↓) | | | | Median MIS (%, ↓) | | | |
|---|---|---|---|---|---|---|---|---|---|---|---|---|---|---|---|---|
| $\tau$ (Å) | GraphDG | CGCF | ConfGF | DGSM | GraphDG | CGCF | ConfGF | DGSM | GraphDG | CGCF | ConfGF | DGSM | GraphDG | CGCF | ConfGF | DGSM |
| 0.10 | 1.03 | 0.14 | 17.59 | 35.43 | 0.00 | 0.00 | 10.29 | 29.64 | 99.70 | 99.93 | 92.30 | 81.38 | 100.00 | 100.00 | 96.23 | 84.42 |
| 0.20 | 13.05 | 10.97 | 43.60 | 59.29 | 2.99 | 3.95 | 37.92 | 57.60 | 96.12 | 96.91 | 81.67 | 69.19 | 99.04 | 99.04 | 85.65 | 72.10 |
| 0.30 | 32.26 | 31.02 | 61.94 | 72.19 | 18.81 | 22.94 | 59.66 | 73.79 | 87.15 | 89.21 | 73.06 | 63.29 | 94.44 | 94.44 | 77.27 | 67.32 |
| 0.40 | 53.53 | 53.65 | 75.45 | 80.82 | 50.00 | 52.63 | 80.64 | 83.84 | 72.60 | 78.35 | 65.38 | 58.34 | 82.63 | 82.63 | 70.00 | 60.32 |
| 0.50 | 73.33 | 78.05 | 88.49 | 91.49 | 84.21 | 82.48 | 94.13 | 95.92 | 56.09 | 63.51 | 53.56 | 46.31 | 64.66 | 64.66 | 56.59 | 49.76 |
| 0.60 | 88.24 | 94.85 | 97.71 | 98.50 | 98.83 | 98.79 | 100.00 | 100.00 | 40.36 | 44.82 | 34.78 | 27.51 | 43.73 | 43.73 | 35.86 | 23.52 |
| 0.70 | 95.93 | 99.05 | 99.52 | 99.58 | 100.00 | 100.00 | 100.00 | 100.00 | 27.93 | 29.64 | 21.00 | 15.52 | 23.38 | 23.38 | 15.64 | 8.05 |
| 0.80 | 98.70 | 99.47 | 99.68 | 99.74 | 100.00 | 100.00 | 100.00 | 100.00 | 19.15 | 20.98 | 12.86 | 9.48 | 10.72 | 10.72 | 5.23 | 0.85 |
| 0.90 | 99.33 | 99.50 | 99.77 | 99.86 | 100.00 | 100.00 | 100.00 | 100.00 | 12.76 | 16.74 | 8.98 | 7.13 | 3.65 | 3.65 | 1.53 | 0.00 |
| 1.00 | 99.48 | 99.50 | 99.86 | 99.94 | 100.00 | 100.00 | 100.00 | 100.00 | 8.00 | 14.19 | 6.76 | 5.81 | 0.47 | 0.47 | 0.36 | 0.00 |
| 1.10 | 99.51 | 99.51 | 99.91 | 99.98 | 100.00 | 100.00 | 100.00 | 100.00 | 4.99 | 12.26 | 5.57 | 5.06 | 0.00 | 0.00 | 0.00 | 0.00 |
| 1.20 | 99.51 | 99.51 | 99.94 | 99.99 | 100.00 | 100.00 | 100.00 | 100.00 | 2.95 | 9.68 | 3.48 | 3.26 | 0.00 | 0.00 | 0.00 | 0.00 |
| 1.30 | 99.51 | 99.51 | 99.96 | 100.00 | 100.00 | 100.00 | 100.00 | 100.00 | 1.65 | 7.48 | 1.91 | 1.73 | 0.00 | 0.00 | 0.00 | 0.00 |
| 1.40 | 99.51 | 99.51 | 99.96 | 100.00 | 100.00 | 100.00 | 100.00 | 100.00 | 0.84 | 5.94 | 1.05 | 0.91 | 0.00 | 0.00 | 0.00 | 0.00 |
| 1.50 | 99.52 | 99.51 | 99.97 | 100.00 | 100.00 | 100.00 | 100.00 | 100.00 | 0.41 | 4.94 | 0.70 | 0.55 | 0.00 | 0.00 | 0.00 | 0.00 |

Table 7: COV and MIS scores of different methods on GEOM-Drugs dataset at different thresholds ($\tau$). ($\uparrow$) indicates higher is better. ($\downarrow$) indicates lower is better.

| Drugs | Mean COV (%, ↑) | | | | Median COV (%, ↑) | | | | Mean MIS (%, ↓) | | | | Median MIS (%, ↓) | | | |
|---|---|---|---|---|---|---|---|---|---|---|---|---|---|---|---|---|
| $\tau$ (Å) | GraphDG | CGCF | ConfGF | DGSM | GraphDG | CGCF | ConfGF | DGSM | GraphDG | CGCF | ConfGF | DGSM | GraphDG | CGCF | ConfGF | DGSM |
| 0.25 | 0.00 | 0.06 | 0.17 | 0.29 | 0.00 | 0.00 | 0.00 | 0.00 | 100.00 | 99.99 | 99.97 | 99.90 | 100.00 | 100.00 | 100.00 | 100.00 |
| 0.50 | 0.26 | 0.80 | 1.15 | 3.85 | 0.00 | 0.00 | 0.00 | 0.00 | 99.95 | 99.80 | 99.52 | 98.34 | 100.00 | 100.00 | 100.00 | 100.00 |
| 0.75 | 0.75 | 5.81 | 9.15 | 19.94 | 0.00 | 0.00 | 0.50 | 8.15 | 99.69 | 97.86 | 96.94 | 91.93 | 100.00 | 100.00 | 99.75 | 97.09 |
| 1.00 | 2.39 | 24.67 | 30.60 | 48.71 | 0.00 | 11.81 | 18.89 | 47.15 | 99.14 | 90.82 | 89.63 | 78.38 | 100.00 | 96.50 | 95.58 | 86.07 |
| 1.25 | 8.27 | 53.96 | 62.15 | 78.73 | 0.00 | 57.06 | 70.93 | 94.39 | 97.92 | 78.32 | 76.58 | 59.92 | 100.00 | 86.28 | 84.48 | 62.85 |
| 1.50 | 19.96 | 79.37 | 86.62 | 93.75 | 4.00 | 92.46 | 98.79 | 100.00 | 94.40 | 63.80 | 60.06 | 42.02 | 99.14 | 66.39 | 63.81 | 37.36 |
| 1.75 | 36.86 | 91.47 | 96.53 | 99.01 | 26.58 | 100.00 | 100.00 | 100.00 | 87.68 | 49.72 | 43.63 | 27.61 | 95.83 | 47.09 | 41.72 | 18.66 |
| 2.00 | 55.79 | 96.73 | 98.62 | 99.90 | 55.26 | 100.00 | 100.00 | 100.00 | 76.99 | 37.53 | 29.80 | 18.04 | 87.35 | 30.90 | 22.44 | 9.23 |
| 2.25 | 71.43 | 99.05 | 99.83 | 99.99 | 80.00 | 100.00 | 100.00 | 100.00 | 61.76 | 27.30 | 18.68 | 12.04 | 69.74 | 20.07 | 10.93 | 6.00 |
| 2.50 | 83.53 | 99.47 | 100.00 | 100.00 | 95.45 | 100.00 | 100.00 | 100.00 | 44.32 | 18.97 | 11.09 | 8.19 | 42.96 | 12.33 | 3.31 | 4.12 |
| 2.75 | 91.09 | 99.60 | 100.00 | 100.00 | 100.00 | 100.00 | 100.00 | 100.00 | 27.92 | 12.52 | 6.32 | 5.62 | 16.67 | 6.82 | 0.74 | 3.04 |
| 3.00 | 95.00 | 99.96 | 100.00 | 100.00 | 100.00 | 100.00 | 100.00 | 100.00 | 15.97 | 7.67 | 3.36 | 3.57 | 2.46 | 3.32 | 0.00 | 2.00 |

## C.5 Ablation Studies

To examine the benefit of dynamic non-bonded edges, we trained another model without non-bonded edges and report its performance in Table 8. We find that when non-bonded edges are removed from our model, the performance drops to a level similar to ConfGF. This confirms the effectiveness of non-bonded edges.

We additionally trained two models with cutoff distances of 5Å, and 20Å on the GEOM-Drugs dataset, and report their performance in Table 9. We find that as the cutoff distance increases (from 5Å to 20Å), the coverage score (COV) decreases. This is because a longer cutoff will lead to more non-bonded edges and introduce additional redundancies, thus reducing the flexibility and the diversity of the generation. The matching score (MAT) slightly increases as the cutoff distance gets longer. We find that the network produces less confident and less accurate predictions for distant interactions and thus affects the generation quality.

Table 8: We trained another model without non-bonded edges on the GEOM-Drugs dataset, and report the COV and MAT scores in the table.

| | COV (%, ↑) | | MAT (Å, ↓) | |
|---|---|---|---|---|
| | Mean | Median | Mean | Median |
| ConfGF | 62.15 | 70.93 | 1.1629 | 1.1596 |
| w/o non-bonded ($\delta = 0$Å) | 68.96 | 79.48 | 1.1524 | 1.1370 |
| **w/ non-bonded** | **78.73** | **94.39** | **1.0154** | **0.9980** |

Table 9: We trained two models with cutoff distances of 5Å, and 20Å respectively on the GEOM-Drugs dataset, and report the COV and MAT scores in the table.

| $\delta$ | COV (%, ↑) | | MAT (Å, ↓) | |
|---|---|---|---|---|
| | Mean | Median | Mean | Median |
| 0Å(w/o non-bonded) | 68.96 | 79.48 | 1.1524 | 1.1370 |
| 5Å | **81.27** | **94.62** | **0.9855** | **0.9697** |
| 10Å | 78.73 | 94.39 | 1.0154 | 0.9980 |
| 20Å | 77.23 | 88.06 | 1.0096 | 1.0118 |

## C.6 Additional Generated Samples

We visualize more generated conformations by our DGSM of different molecular systems, i.e., drug-like molecules, multi-molecular complexes and protein sidechains, in Figure 7.

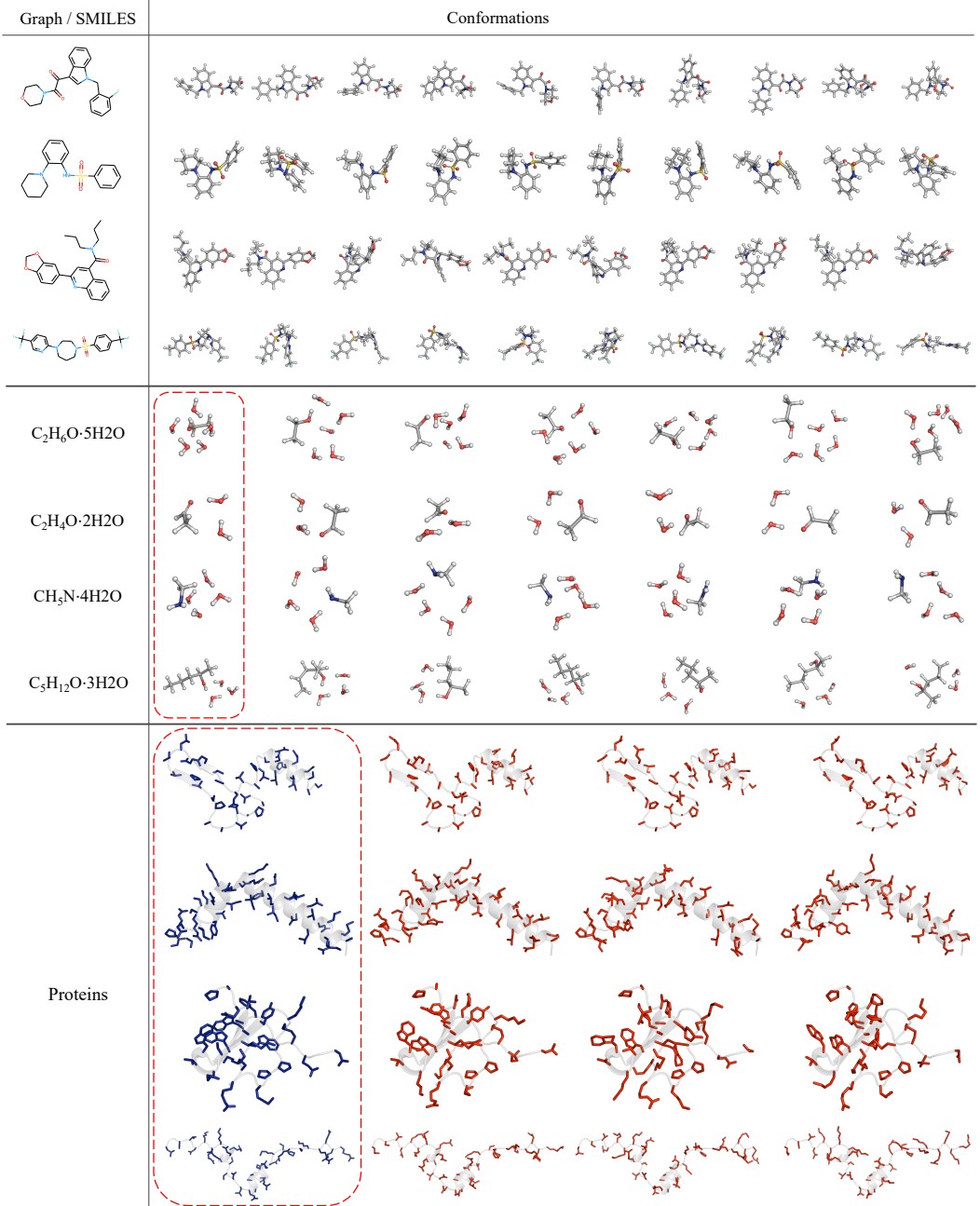

Figure 7: Visualization of conformations generated by DGSM. The reference conformations are highlighted by red dashes. For protein sidechains, the ground-truth sidechain (blue) and the generated sidechain (red) are highlighted.