# OpenReview forum: "Predicting Molecular Conformation via Dynamic Graph Score Matching"
_NeurIPS.cc/2021/Conference — NeurIPS 2021 Poster_

### Official Review · Reviewer_fHJN · 2021-07-14

**Rating:** 7
**Confidence:** 3

**Summary:**

This paper addresses the task of molecular conformer generation, i.e. generate the 3D conformer ensemble conditioned on the molecular graph. The introduced method, Dynamic Graph Score Matching (DGSM), is using score matching to generate 3D structures by explicitly capturing both local and long-range interactions. This is done using dynamic graphs constructed via atom coordinate perturbations using different noise levels as required by the denoising score matching procedure.  The score is estimated using message passing neural networks which operate on these dynamic graphs. At inference time, Langevin dynamics is used to sample conformations from the generated distribution. Empirically, the authors show clear improvements over previous machine learning (ML) models and over one popular open-source non-ML method (RDKit) on molecular conformer generation on two different tasks, as well as improvements on experiments on property prediction based on molecular conformations, protein sidechain conformation generation, and multi-molecular complex conformation.

In short, I think this approach is quite compelling and would be very impactful for future work related to graph conditional 3D generative models of various data in biology, chemistry, etc. The paper is very well written and the method is clean, addressing nicely using dynamic graphs the very important problem of modeling long-range interactions in 3D generation.

**Limitations And Societal Impact:**

Limitation section is missing.

**Main Review:**

1. Originality:

Using denoising score matching for 3D conformer generation is a novel and non-trivial contribution.  The authors employ dynamic graphs for this problem and describe all challenges and solutions. Related work is well written, but it might be useful to discuss non-ML literature a bit more in depth, e.g. see [1].


2. Quality:

I already argued above that this is a good quality submission. However, the authors do not discuss any limitations of their work. It would be great if they could describe common failure cases of their conformer generation model. Also, how stable is the training of the denoising score matching procedure ?

3. Clarity: paper is well written.

4. Significance: already commented above. This paper provides a nice clean solution to a very important problem in graph conditional 3D generation.


5. Improvement points: while I generally like this work, there are a few points where the authors could improve their paper.

- experimental results in table 1: COV and MAT only show information about the recovered ground truth conformers, i.e. equivalent of recall. It would be also very useful to understand how many of the generated structures are close to ground truth conformers, i.e. to show the equivalent metrics of precision. Also, including a stronger non-ML baseline (e.g. OMEGA https://www.eyesopen.com/omega ) would be helpful in making this work more visible and useful in the chemistry community.

- it is known that conformations of a single molecule are usually changing mostly in terms of their rotatable bonds. Can the authors comment on how torsion angles can be integrated in their framework ?

- chirality is a very important issue for molecular conformer generation that was not explicitly tackled until very recently (see [2]). Can the authors comment on whether this can be integrated in DGSM?


[1] Conformation Generation: The State of the Art, Hawkings, 2017
[2] GeoMol: Torsional Geometric Generation of Molecular 3D Conformer Ensembles, Ganea et al., 2021

**Time Spent Reviewing:**

3

---

> ### Author Response · Authors · 2021-08-10
> **Response to the Reviewer fHJN**
>
> Thanks for your comments and suggestions. We will discuss the limitations of the work in our subsequent revision. The response to your concerns are listed below:
>
> **[Q] Suggestion on adding new metrics**
>
> Very good point! Following the previous works [1], currently, we use the mean and median COV and MAT scores to evaluate the quality and the diversity of generated conformations. We will consider adding a new metric and a non-ML based baseline in our revision.
>
> **[Q] handling torsional angles and the chirality (Stereochemistry)**
>
> This is a very good point! Currently, the design of DGSM is insensitive to torsional angles and stereochemistry. However, our DGSM is actually very general and can be extended to handle stereochemistry by predicting the gradients of logarithm density of torsional angle with respect to coordinates, i.e.,$\nabla_x[ \log p\theta(x) ]$, where $\theta$ is a function calculating torsional angle from coordinates and x is the coordinates. Note that torsional angles can be calculated in close-form from 3D coordinates and are also invariant to rotation and translation. The chirality can be characterized by the torsional angle among related atoms. We will add the discussion about chirality in our subsequent revision.
>
> **[Q] common failure cases and training of the denoising score matching procedure**
>
> The training of the denoising score matching procedure is quite stable and the loss decreases steadily. So far we have not noticed any common failure case for conformation generation, but we argue that the hyper-parameters of the Langevin dynamics are crucial for the quality of the generated conformations. Empirically, we find the strategies provided by [2] work well.
>
> *We hope the above response could address your concerns.*
>
> **Reference**
> * [1] Learning neural generative dynamics for molecular conformation generation. ICLR, 2021.
> * [2] Improved techniques for training score-based generative models. NeurIPS, 2020.

---

> > ### Comment · Reviewer_fHJN · 2021-08-19
> > **re**
> >
> > Thanks for the reply.
> >
> > I agree with the other reviewers that the methodological contribution of DGSM w.r.t. ConfGF is relatively small, thus the novelty of DGSM is limited to dynamically adding graph edges for different noise levels, while relying on the existing ConfGF method for the rest of the training/inference procedures. However, modeling long-range interactions is an important problem and this paper shows compelling experimental results.
> >
> > Regarding one of my previous points:
> >
> > - it would have been useful to see precision COV and MAT metrics to assess the quality of all generated conformers given that previous ML methods produce a significant fraction of poor conformations, e.g. for Geom-DRUGS, as showed in [a]. This is especially important given that DGSM samples 2x the number of ground truth conformers.
> >
> > [a] GeoMol: Torsional Geometric Generation of Molecular 3D Conformer Ensembles, Ganea et al., 2021

---

> > > ### Author Response · Authors · 2021-08-19
> > > **Reply**
> > >
> > > **[Q] precision COV and MAT metrics**
> > >
> > > We agree with you that the **precision COV and MAT metrics** are important to assess the quality of the generated conformations, and thanks for pointing out the GeoMol paper.
> > >
> > > Note that the GeoMol paper is available on arxiv after the paper submission deadline of NeurIPS. We will definitely cite this paper and include the experimental results on **precision COV and MAT metrics** in our next revision.
> > >
> > > **[Q] novelty**
> > >
> > > Please refer to our response to the Reviewer ssuS

---

### Official Review · Reviewer_ssuS · 2021-07-15

**Rating:** 6
**Confidence:** 4

**Summary:**

This paper proposes to generate molecular conformation by score matching. The proposed method, called DGSM, learns a score network to predict gradient field with respect to a predicted conformation. The score network is parameterized as a message passing network over a dynamic graph between atoms. The graph is constructed based on pairwise distance between atoms in the current conformation. Finally, DGSM uses the learned gradient field along with Langevin dynamics to sample molecular conformations at test time. DGSM is evaluated on two conformation generation benchmarks and achieved new state-of-the-art results.

**Limitations And Societal Impact:**

This paper did not address its limitations. I have included constructive feedback in the review.

**Main Review:**

Overall, this paper is a nice (but small) extension to ConfGF (Shi et al., 2021), which first used score matching methods to generate molecular conformations. DGSM adopts the same score matching framework in ConfGF but modifies the score network architecture. The reviewer acknowledges the importance of dynamic graph construction and the experimental results show promising results.

### Strength

* An end-to-end framework for molecular conformation generation. The method is mostly data driven and does not use much domain-specific rules.
* Experimental results indeed shows that DGSM outperforms ConfGF, supporting the main claim of the paper.
* New experiments on macromolecules and multi-molecule complexes are interesting.

### Weakness
* DGSM is a small extension to ConfGF. The score matching framework has been proposed in prior literature, so the originality of DGSM is relatively small.
* There is no quantitative results on multi-molecule complex conformation. What's the performance of ConfGF for this task?
* The multi-molecule complex task seems artificial. Why not try your algorithm on protein-protein complex conformation prediction?

**Time Spent Reviewing:**

1.5 hours

---

> ### Author Response · Authors · 2021-08-10
> **Response to the Reviewer ssuS**
>
> Thanks for your comments and suggestions. The response to some of your concerns are listed below:
>
> **[Q] Novelty**
>
> * The main contribution of our work is orthogonal to that of ConfGF’s, though it is quite related to our work.
> * In this work, we focus on tackling the long-range non-bonded interaction in molecular conformation generation, which is overlooked in all previous ML-based methods [1,2,3] for this task. Built upon score-based generative models, the proposed DGSM models both the local and the long-range interactions by dynamically constructing graph structures during both training and inference.
> * We argue that modeling long-range interactions in conformation generation not only improves the model’s performance, but also empower the model to handle macro-molecular and multi-molecular systems, which represent a broader range of problems.
>
> **[Q] ConfGF for multi-molecular complexes.**
>
> ConfGF and other ML baselines only model the interactions between bonded atoms within k-hop (k=3 typically) on the molecular graphs. However, in multi-molecular complexes, there are no bonds between different molecules. Therefore, ConfGF is not applicable to multi-molecular complexes.
>
> **[Q] Larger systems such as protein-protein complexes.**
>
> * The experiment on multi-molecular complexes is a proof-of-concept, aiming at demonstrating our model’s capability of solving a broader range of problems. As we have demonstrated that our model is capable of modeling multi-molecular complexes, we believe our model has the potential in modeling protein-protein complexes within the proposed DGSM framework.
> * The current DGSM mainly focuses on inter-atomic interactions. However, to better model proteins, one should at least augment the framework to incorporate inter-amino-acid interactions, which is beyond the consideration of this work. We believe it is valuable to explore how to apply DGSM to proteins by modeling not only inter-atomic but also inter-amino-acid interactions, and we leave it as a future work.
>
> *We hope the above response could address your concerns.*
>
> **Reference**
> * [1] A generative model for molecular distance geometry. ICML 2020.
> * [2] Learning neural generative dynamics for molecular conformation generation. ICLR, 2021.
> * [3] Learning gradient fields for molecular conformation generation. ICML 2021.

---

### Official Review · Reviewer_d6Qw · 2021-07-16

**Rating:** 7
**Confidence:** 5

**Summary:**

The authors develop an improved method for 3D molecular conformation generation from 2D molecular graphs that can more directly incorporate non-bonded, topologically long-range interactions into the generation process. Their starting point is a recent method that leverages score matching with a GNN based on the 2D molecular graph topology, or more specifically, a GNN that is trained to estimate the gradient of the noise-perturbed log density of interatomic distances on the graph edges. They follow the same framework, but add additional edges to the graph topology based on the current perturbed distances when they fall below a certain cutoff (10 Angstroms). Through their experiments, they show that this relatively modest modification signifcantly improves generation performance of small molecule conformers and can be applied to larger and more complex systems such as protein side chain packing and multi-molecular complexes.

**Limitations And Societal Impact:**

I would suggest that the authors explicitly discuss limitations of the model with respect to chirality. Since the model is based purely on distance-based score matching, it will not be able to differentiate between stereoisomers. I imagine that this is likely already causing problems for protein side chain prediction, since some amino acid side chains, such as isoleucine and threonine, contain chiral carbons.

**Main Review:**

Strengths

Motivation and Significance.
Non-bonded interactions are central to understanding larger molecular systems but have been a blind spot or weakness for many ML-based approaches to conformation generation, in part because explicitly modeling all possible pairwise interactions is expensive. This work offers a tractable solution to this by using the domain-motivated design decision of spatial neighborhood cutoffs. This very nicely combines with score matching based generative modeling, since with that framework, you can essentially train on static randomly sampled graphs with approximately linear scaling in the number of atoms, but you still retain the power to dynamically account for arbitrary ij interactions at sampling time.

Small molecule results.
The results on the GEOM datasets are quite strong and already show that non-bonded interactions can make a difference for small molecular systems such as QM9 and drug-like molecules. The improved typical coverage and accuracy seem like qualitative improvements that could make a meaningful impact on conformer generation in molecular property prediction or in-silico discovery efforts. Additionally the improved ensemble energetics from a QM method in Table 2 suggest that this framework is improving the ability of ML-based conformer methods to model these systems in a physically accurate way.

Clarity.
I found the paper well written and easy to follow, with effective figures and notation.

Weaknesses

Larger system baselines.
Both the protein side chain and multi-molecular complex experiments suggest potential applications of the framework, but it would significantly strengthen the paper to add some kind of external baseline model performance to calibrate expectations. For example, although the authors mention that rotamer based side-chain predictors perform much better, it would be helpful to add a baseline such as Rosetta to understand how far the model is from being practically applicable in that domain.

Novelty and ablation.
The primary methodological change here is simple, but I don't that think that is a big weakness because it has clear and meaningful impacts on the capability of the framework. Since that change is so specific, though, I think that it would benefit the paper for the authors to more thoroughly investigate the role of the cutoff distance. For example, does making the cutoff distance small add a useful inductive bias, or is it more a matter of a performance/speed tradeoff?

---Update post-response---
Thank you for adding the additional results about the role of the cutoff distances and how it affects both accuracy and diversity. It is interesting that, at least in current form, there can be benefits to smaller neighborhoods. The authors' point about the method being a general framework is well taken and also the main reason I did and would recommend acceptance (and keep my score the same). I agree it will be quite interesting to see how it can compare to application-specific methods with additional domain-specific tuning.

**Time Spent Reviewing:**

4

---

> ### Author Response · Authors · 2021-08-10
> **Response to the Reviewer d6Qw**
>
> We thank the reviewer for the constructive comment. We will discuss the limitations of the work in our subsequent revision, especially the issue related to stereochemistry. Below is our response to the reviewer’s concerns:
>
> **[Q] The role of the cutoff distance**
>
> We will present more experiments with different cutoff distances and different datasets in our subsequent version. For now, we present two additional models with cutoff distances at 5A, and 20A on the GEOM-Drugs dataset. Below are the COV and MAT scores:
>
> |         | **COV (Mean)** | **COV (Median)** | **MAT (Mean)** | **MAT (Median)** |
> | ------- | :------------: | :--------------: | :------------: | :--------------: |
> | **5Å**  |   **81.27**    |    **94.62**     |   **0.9855**   |    **0.9697**    |
> | **10Å** |     78.73      |      94.39       |     1.0154     |      0.9980      |
> | **20Å** |     77.23      |      88.06       |     1.0096     |      1.0118      |
>
> * We find that as the cutoff distance increases (from 5A to 20A), the coverage score (COV) decreases. This is because a longer cutoff will lead to more non-bonded edges and introduce additional redundancies, thus reducing the flexibility and the diversity of the generation.
> * The matching score (MAT) slightly increases as the cutoff distance gets longer. We find that the network produces less confident and less accurate predictions for distant interactions. It might be a factor impacting on the MAT score when the cutoff distance is longer.
> * Overall, the table shows that a cutoff distance of 5A performs the best out of (5A, 10A, 20A). More detailed results (different cutoff distances and different datasets) will be presented in the revision of the paper.
>
> **[Q] Larger system baselines.**
>
> * We thank the reviewer for the constructive suggestion. We will consider adding other sidechain packing baselines in our subsequent version.
> * Also, we would like to kindly point out that our method is a general framework. In our sidechain experiment, we simply treat protein as a set of atoms, like how we treat small molecules. We believe this is by no mean the best way for modeling atom-level protein structures. However, since our method is a general framework, we believe it is valuable to investigate how to incorporate problem-specific knowledge within the framework. For example, modeling not only inter-atomic interactions but also higher level inter-amino-acid interactions for the sidechain structure prediction problem.
>
> *We hope the above response could address your concerns.*

---

### Official Review · Reviewer_S9Sr · 2021-07-24

**Rating:** 6
**Confidence:** 3

**Summary:**

This paper introduces a new approach for generating molecular conformations, using score-based generative modeling.  The key contribution over prior work is the dynamic addition (during both training and inference) of long-range interactions between non-bonded atoms that are not proximal in the molecular graph.  As these interactions are often critical to determining the underlying molecular conformations, the authors contend that their addition leads to improved performance on a range of tasks.  In particular, they demonstrate improved performance over previous state-of-the-art on the generation of conformations for small molecules, protein side chains, and multi-molecular complexes, as well as small-molecule ensemble property prediction.

**Ethical Concerns:**

None.

**Limitations And Societal Impact:**

The authors mostly do not discuss limitations of this work.  What size systems can currently be modeled?  The use of other computational methods (as opposed to direct experimental measurements) to provide ground truth data should also be addressed.

I do not believe there are any potential negative societal impacts to be addressed here.

**Main Review:**

Originality: While score-based generative modeling has been recently applied to molecular conformation generation (ref 32), and 3D information has been incorporated into many molecular learning algorithms, the dynamic addition of non-bonded atoms within the score matching paradigm is novel.  Most of the tasks are not new, with the exception of the multi-molecular complex conformation task (which does look quite interesting!).

Quality: This is my main concern regarding this work.  I believe further experiments are required to conclusively support the claim that DGSM's improved performance derives from taking into account long-range interactions.  I realize that at a high-level CONFGF is DGSM without these non-bonded edges, though it is also unclear if there are other methodological differences that could be coming in to play.  In particular, I would like to see an ablation study comparing DGSM's performance with and without the augmentation of the original graph structure with non-bonded edges.  Even better would be plots demonstrating the change in performance of DGSM across different cutoff thresholds $\delta$ (currently fixed to 10 Å).  I in fact would expect that these plots might look different across tasks involving small molecules, proteins, and multi-molecular complexes.

Clarity: The work is clearly written and well organized.

Significance: The results are overall quite compelling.  It is unclear if there will be much use (yet) of this method for larger molecular systems and complexes since specialized methods that incorporate a lot of domain knowledge are often state-of-the-art.

*** Post-response update:

The reviewer addressed my concern regarding the non-bonded edge ablation, and the results support the overall conclusion, so I am raising my score.  I do think the decrease in performance as the cutoff distance increases is an interesting line of investigation to be pursued further.

**Time Spent Reviewing:**

5

---

> ### Author Response · Authors · 2021-08-10
> **Response to the Reviewer S9Sr**
>
> Thank you very much for your constructive comments! We have conducted more experiments according to your suggestions, and we will discuss the limitations of the work in our subsequent revision. The response to some of your concerns are listed below:
>
> **[Q] Effectiveness of non-bonded edges**
>
> We trained another model without non-bonded edges on the GEOM-Drugs dataset. The COV and MAT scores are listed below:
>
> |                    | **COV (Mean)** | **COV (Median)** | **MAT (Mean)** | **MAT (Median)** |
> | ------------------ | :------------: | :--------------: | :------------: | :--------------: |
> | **ConfGF**         |     62.15      |      70.93       |     1.1629     |      1.1596      |
> | **w/o non-bonded** |     68.96      |      79.48       |     1.1524     |      1.1370      |
> | **w/ non-bonded**  |   **78.73**    |    **94.39**     |   **1.0154**   |    **0.9980**    |
>
> * We find that when non-bonded edges are removed from our model, the performance drops to a level similar to ConfGF, thus confirming the effectiveness of non-bonded edges.
> * We would also like to kindly remind the reviewer to take a closer look at Figure 4, where the best predictions from different models are presented. In the second row of the figure, we notice that there are no clashes (too close) between non-bonded atoms, indicating that considering long-range interaction avoids unrealistic substructures such as clashes, and thus improves the prediction performance. We will add the full results in the revised version.
>
>
> **[Q] Influence of different cut-off thresholds**
>
> For now, we additionally trained two models with cutoff distances of 5A, and 20A respectively on the GEOM-Drugs dataset. Below are the COV and MAT scores:
>
> |         | **COV (Mean)** | **COV (Median)** | **MAT (Mean)** | **MAT (Median)** |
> | ------- | :------------: | :--------------: | :------------: | :--------------: |
> | **5Å**  |   **81.27**    |    **94.62**     |   **0.9855**   |    **0.9697**    |
> | **10Å** |     78.73      |      94.39       |     1.0154     |      0.9980      |
> | **20Å** |     77.23      |      88.06       |     1.0096     |      1.0118      |
>
> * We find that as the cutoff distance increases (from 5A to 20A), the coverage score (COV) decreases. This is because a longer cutoff will lead to more non-bonded edges and introduce additional redundancies, thus reducing the flexibility and the diversity of the generation.
> * The matching score (MAT) slightly increases as the cutoff distance gets longer. We find that the network produces less confident and less accurate predictions for distant interactions. It might be a factor impacting on the MAT score when the cutoff distance is longer.
> * Overall, the table shows that a cutoff distance of 5A performs the best out of (5A, 10A, 20A). We will present more experiments with different cutoff distances (e.g., less than 5A) and different datasets in our subsequent revision.
>
> **[Q] Question on the size of the systems that DGSM can model and the use of computational methods for ground-truth data.**
>
> * Currently we apply the DGSM to small molecules, multi-molecular complex and protein sidechains, whose size range from dozens of atoms (small molecules) to thousands of atoms (protein with atomic resolution). Since both the training and inference of DGSM is efficient, we believe it can be applied to more large-scale systems, e.g., protein-protein interaction complex.
> * For the use of computational methods, the official GEOM dataset [1] use xTB and CREST, the sidechainnet is mined from PDB database, and the multi-molecular complex data uses xTB.
>
> *We hope the above response could address your concerns.*
>
> **Reference**
> * [1] GEOM: Energy-annotated molecular conformations for property prediction and molecular generation

---

### Decision · Program_Chairs · 2021-09-27

**Decision:**

Accept (Poster)

**Comment:**

The reviewers all agreed this work should be accepted. They appreciated the clarity, the results (particularly the small molecule results), the motivation of the problem being addressed, the end-to-end aspect of the model, and the related work section. The authors should take into account the detailed suggestions of the reviewers for the camera ready version, to make an already nice paper even better.